# How does digital transformation improve new product development performance from the perspective of resource orchestration?— Analysis based on configuration

**Daohai Zhang**[1]*, **Xianqiao Hou**[1], **Jiayuan Guo**[2]

**1** School of Management, Jiangsu University, Zhenjiang, Jiangsu, **2** School of Business, Hohai University, Nanjing, Jiangsu

* zdh@ujs.edu.cn

**Data Availability Statement:** All data sources are public annual reports of Chinese listed companies, which can be reorganized by individuals: http://www.sse.com.cn/ http://www.szse.cn/disclosure/listed/fixed/index.html http://www.cninfo.com.cn/

## Abstract

The new product development (NPD) activities of enterprises serve as a critical source of competitiveness. To effectively harness the opportunities presented by digital transformation and enhance the performance of NPD in organizations amidst the digital revolution is an area of concern that warrants attention. In this context, we conducted research using data from the annual reports of 35 listed mainboard enterprises in 2021. This research used the resource arrangement theory and the resource-structure-capability research framework. In addition, we utilized the quantitative comparative analysis (QCA) method to investigate how digital transformation capability, R&D investment capability, and heterogeneity synergies impact the performance of NPD. The findings indicate that: (1) Four distinct paths (i.e., digital innovation-driven, large-scale multi-talent, mature and robust, and digital start-up) drive improvements in NPD performance. Notably, there exists an asymmetric causal relationship between these four paths and the performance; (2) Digital transformation capability, firm R&D investment, and firm heterogeneity all contribute to enhancing NPD performance. However, they do not individually guarantee high performance. A synergistic effect of at least two factors is required to yield notable NPD performance; (3) Enterprise heterogeneity plays a pivotal role. Companies with different characteristics must opt for distinct digital transformation paths to improve their NPD performance; (4) In the initial stage of digital transformation, enterprises can enhance NPD performance by augmenting their investment in R&D personnel.

## Introduction

New product development (*hereafter called NPD*) is the process by which enterprises transform market opportunities and product technologies into new products and commercialize them through a series of technical activities [1]. According to Schumpeter's theory of innovation, NPD is a crucial approach for companies to adapt to changing market and technological

[new/index](new/index) Original data can be provided through personal email.

**Funding:** The author(s) received no specific funding for this work.

**Competing interests:** The authors have declared that no competing interests exist.

conditions [2]. It constitutes a vital aspect of manufacturing enterprises, and a significant means for them to gain competitive advantage [3]. The success or failure of NPD directly impacts whether an enterprise can thrive in the fiercely competitive market environment [2]. However, in the current era of the digital wave, the continuous development of big data technology has disrupted the traditional profit channels and business models for enterprises. The business landscape has grown increasingly challenging, with information sharing reducing the information gap between enterprises and diminishing the disparity in external resources they can access. Thus, there is growing need to prioritize the ongoing advancement of internal organization informatization, the integration of enterprise resources, and the enhancement of long-term competitiveness. Simultaneously, as technology progresses, product life cycles are shortening, increasing the pressure on enterprises engaged in NPD to continuously meet consumer demands. To secure a foothold in the dynamic and ever-changing market, enterprises must continually introduce new products and enhance their NPD performance.

The success or failure of NPD is influenced by numerous factors. Previous research on NPD performance has predominantly examined it from the following dimensions. From an environmental dimension, the NPD process encounters many challenges with environmental changes representing a significant hurdle [4]. Strong environmental fluctuations often act as the primary catalyst for enhancing existing NPD processes [5]. Gong Yanping et al. [6] delved into the effectiveness of R&D integration mechanisms in response to environmental dynamics. Within the organizational dimension, the ability to learn within the organization has a direct impact on the absorption and assimilation of knowledge [7]. The organization's capacity to absorb, digest, and apply knowledge contributes to improved NPD performance [8]. Additionally, some scholars argue that an organization's ability to forget certain knowledge can positively influence NPD performance [9, 10]. In the project dimension, knowledge resources, as a project-level factor play a significant role in NPD [11]. Research on knowledge resources primarily encompasses three aspects: knowledge integration, knowledge management, and knowledge sharing. In particular, knowledge integration fosters knowledge circulation within an enterprise. The recombination of knowledge enhances an enterprise's dynamic capabilities, enhances the competitiveness of newly developed products, and improves NPD performance [12]. Knowledge Management lies at the Heart of NPD [11–13], and knowledge sharing also exerts a positive effect on NPD [14]. The external relationship dimension revolves around the connection between external factors and the enterprise. Li [15]argues that customer involvement contributes to the success of NPD, and Yao Shanji et al. [16] argue that customer participation not only affects the speed of new product launches, but also influences the degree of new product innovation. Wang Tao et al. [17] suggest that customer dependence promotes customer participation in NPD.

In today's landscape, the digital economy is continuously evolving. With the support of initiatives like "Made in China 2025" and other national strategies, China's manufacturing enterprises have entered a new phase of rapid and high-quality development in their digital transformation journey. The report from the *Party's Twentieth National Congress* emphasizes high-quality development as a primary goal in building a socialist modernization in the country. As a crucial avenue for achieving high-quality development in manufacturing enterprises, enterprise digital transformation has become a widely recognized and researched topic. Numerous studies on digital transformation and enterprise performance have demonstrated its association with various aspects such as innovation and financial performance [18–20]. However, there has been relatively less research on the relationship between digital transformation and NPD performance. In the existing studies, Chi Maomao et al. [21] initiated their study with a focus on concept of digital empowerment. The investigation aimed to explore the impact of digital transformation and R&D capability on performance of NPD. In a similar

vien, Shan Biaoan et al. [22] posited that performance of NPD is influenced by enterprise digital transformation and internal factors. Thus, it evident that digital transformation is not a singular factor that impacts NPD performance alone.

Unlike previous research that focused the on the four dimensions of NPD, advanced digital technologies have considerably altered the manner in which contemporary businesses develop and compete, thereby providing new methods for enterprise value creation. In the context of new technology and era, enterprise digital transformation profoundly influences both internal and external aspects of companies. Particularly, when enterprises face bottlenecks in their development, digital transformation can drive strategic changes, reshape the way they create value, and enhance their overall performance [23]. Therefore, the key for enterprise lies in how they seize the opportunities and address the challenges presented by the digital economy revolution. It entails harnessing the positive potential of digital transformation elements, optimizing the utilization of limited resources and achieving breakthroughs in their respective industries.

Based on a configuration perspective and utilizing the fuzzy-set qualitative comparative analysis (fsQCA) method, this study investigates the performance of NPD in manufacturing enterprises under the digital background. Specifically, the study analyzes the synergistic effect and complex interaction of the influencing factors and further elucidates the influence mechanism of digital transformation on NPD performance. The paper's contributions are threefold. Firstly, by adopting a configuration perspective and focusing on a country listed manufacturing enterprises, this study delves into the impact of six conditional combinations on NPD performance at three levels: ability, resources, and structure. This approach enhances our comprehension of the intricate interactions among multiple factors that influence NPD performance. Additionally, it offers valuable insights for deepening our understanding of the complexities behind NPD performance. Moreover, it provides valuable insights for driving the development of China's manufacturing industry and enhancing enterprise competitiveness. Secondly, by incorporating the heterogeneous nature of enterprises into the research framework, this paper enriches the study of enterprise performance scenarios. This enrichment contributes to elevating digital capabilities, enhancing NPD abilities, and boosting the competitive advantages of various enterprises. Thirdly, the data of this study are all from the annual reports of enterprises, which are real and credible, and the analysis results are highly persuasive. This study not only helps enterprises to better understand the mechanism of resource management effectiveness in the NPD process, but also facilitates enterprise managers to formulate effective resource management plans to improve their NPD.

## Theoretical basis and research framework

Resource orchestration theory emphasizes that the resources alone do not spontaneously appear and create value for enterprises. Rather, it is the actions taken by the enterprise with these resources that generate value and secure a competitive position in the industry. This perspective encompasses the process of resource orchestration, which includes resource construction, resource bundle collection, and resource leveraging [24], as well as the role of resource orchestration involving human resources, capital, strategic orientation, and more [25]. Drawing from this, Huang Hao et al. [26] developed a theoretical model known as "Resource arrangement-entrepreneurial capability-Growth Performance", suggesting that both resources and capabilities contribute to enterprise growth. Wang Lin et al. [27] introduced the theoretical analysis framework of "Resource-Structure-Capability", and established a value co-creation mechanism based on the resource-based view to enhance the capacity for value acquisition. Li Wen et al. [28] constructed a theoretical model of business model innovation by integrating

the enterprise network (resource layer) and big data's ability (ability layer). In summary, the perspective of resource arrangement underscores the significance of resource management, utilization, and transformation in enhancing enterprise performance.

Thus, grounded on the tenets of the resource scheduling theory, this study focuses on manufacturing enterprises listed on the main board as its research subjects. This paper delves into the complexity and manifold aspects of NPD from a tripartite aspects, namely: digital transformation capabilities, research and development input, and heterogeneity of firms.

## Digital transformation capabilities (Capability layer)

New generation information technologies, including artificial intelligence, virtual reality / augmented reality, cloud computing and 5G, lay the foundation for the digital transformation of enterprises. Bjrkdahl [29] demonstrates that digital transformation significantly enhances production efficiency and facilitate the development more complex product and service system, leading to a more comprehensive value chain.

According to the theory of factor distribution, data, now an independent factor of production, can be viewed as an extension of information technologies (IT's) involvement in factor distribution and pricing. Through digital transformation, companies utilize big data and cloud computing technologies to overcome business barriers, dissolve "data islands", foster the flow of all diverse data types, and enhance corporate performance. Digital transformation empowers enterprises to leverage advanced digital technologies more effectively, generate digital products and optimize operational models [30, 31]. It also drives organizational reform within enterprises to align with the digital economy's evolution, establish digital business models, and elevate digital competitiveness [32, 33].

In line with Xu Xianchun [34] et al.'s research, the broad constitutes of the digital economy encompass a pivotal indicator: the construction of digital infrastructure. This infrastructure serves as the bedrock for guaranteeing the digital economy's functioning and expansion. At its core, the economy employs digital innovations like big data, cloud computing, and the Internet of things to steer the digital transformation [35]. Rogers [36], on other hand, asserts that it's not digital technology alone that influences enterprise digital transformation; rather, it's the digital strategy of enterprises that truly drives this process. The guidance provided by a digital strategy alludes to enterprises' proactive embrace of digital strategies. This, to some extent, reflects the enterprises' commitment and capability to invest in the realm of digitization. Enterprises through the guidance of digital strategy, accomplish digital transformation, innovate through digital technology, and consequently enhance their competitiveness [37]. Therefore, this paper selects **(a)** digital infrastructure construction and **(b)** digital strategy as the conditional variables of digital transformation capability.

## Enterprises' R&D inputs (Resource layer)

From the perspective of resource-based theory, the innovative capabilities that enterprises accumulate during their development, along with the innovative technologies and resources generated through this process, can establish enduring competitive advantages. These unique advantages and capabilities are not easily surpassed or replicated in the short term, ultimately becoming a source of profitability for enterprises [38]. R&D investment plays a pivotal role in promoting technological innovation [39], enhancing the independent innovation capabilities of enterprises, boosting their competitiveness, gaining consumer trust in products, and significantly influencing consumers purchase decisions.

The main conclusions regarding the relationship between R&D investment and firm performance fall into two main categories. The first perspective suggests that R&D investment

positively impacts enterprise performance. Brand [40] found that R&D investment enhances enterprise performance when examining listed U.S. companies. According to the endogenous growth theory, endogenous technological progress is the key to sustaining economic growth. For enterprises, R&D investment affects performance through the cumulative effects of knowledge, technology, capital and economies of scale [41]. The second viewpoint posits a non-linear relationship between R&D investment and firm performance, indicating the potential for both promotion and inhibition. For example, Wang [42] found that open innovation can improve NPD performance, while the intensity of R&D investment by firms might negatively affect NPD performance. Among domestic scholars, Wang Xiquan et al. [43] argue that there exists an inverted U-shaped relationship between enterprise R&D investment and NPD performance. Considering the research mentioned above, it is evident that the impact of R&D investment on enterprise performance is not isolated. Consequently, this paper adopts a comprehensive and systematic analytical approach grounded in set theory to explore the interrelationships between R&D investment and other factors.

R&D investment primarily comprises capital expenditure and manpower investment. Therefore, when studying enterprise's R&D investment, key indicators encompass R&D capital investment and R&D personnel investment [44]. As a result, this paper designates **(c)** R&D capital investment and **(d)** R&D personnel investment as the conditional variables at this level.

## Firm heterogeneity (Structural layer)

In examining the two dimensions mentioned above and their impact on firm performance, it is generally found that firm heterogeneity plays a role in NPD performance, including factors such as firm size and firm age. However, in previous empirical studies, most scholars have treated enterprise heterogeneity as a control variable. The challenge to address is how to effectively promote the digital transformation of enterprises of varying sizes and different ages, leading to enhanced NPD performance. Among existing studies, Sun Zhongjuan et al. [45] employed enterprise size as a threshold variable to investigate the impact of policy support. They concluded that for achieving high-quality policy support, differences among enterprises should be considered, necessitating the implementation of "targeted support". Wen Jun et al. [46] in their study of enterprise R&D investment, introduced enterprise size as a variable and found a significant positive relationship between enterprise size and the variables related to R&D investment, with a notable negative moderating effect. This positive relationship was particularly pronounced in the context of small and medium-sized enterprises.

Although Yang Liuqing et al. [47] incorporated firm characteristics as a control variable, they arrived at the conclusion that a firm's age significantly influences its R&D and production activities. Specifically, the length of the firm's age directly determines the magnitude of sunk costs, thus confirming the hypothesis that a firm's age is significantly negatively correlated with R&D investment. Consequently, the length of the firm's age is a crucial factor impacting R&D investment. As a result, this paper designates **(e)** Size and **(f)** age of the enterprise as the conditional variables of this dimension.

## Theoretical framework

Taking a comprehensive view of the literature on the factors influencing NPD performance mentioned above, this paper identifies six antecedent variables: digital infrastructure, digital strategy, R&D capital investment, R&D personnel investment, enterprise size and enterprise age. These variables are selected from the three dimensions of capability, resource, and structure. The study then employs fsQCA to explore the pathways that enterprises follow to

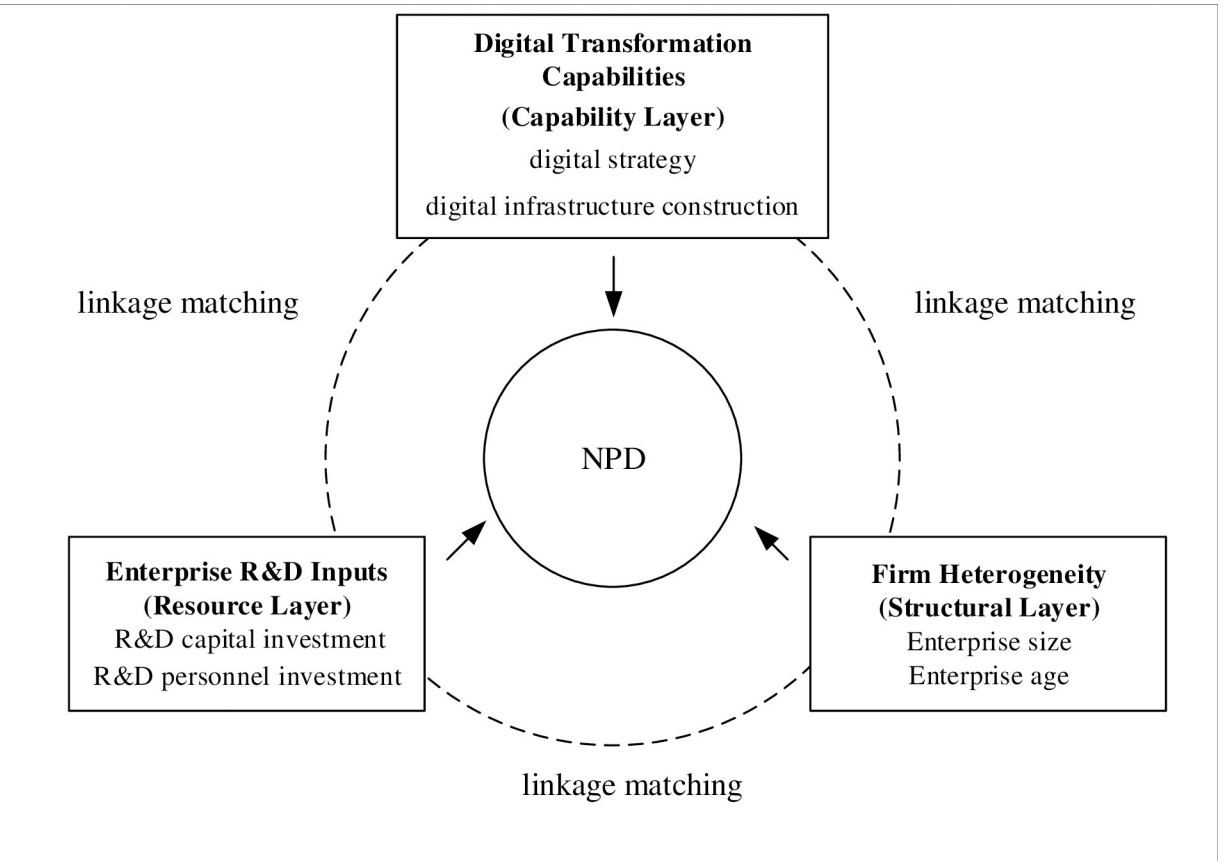

**Fig 1. Theoretical framework.**

improve NPD performance in the context of digitization. The constructed path to improve NPD performance is depicted as the theoretical framework illustrated in Fig 1.

## Methodology

### Quantitative comparative analysis (QCA)

The research used QCA. In comparison to traditional regression analysis, QCA, based on histogram analysis, offers the advantage of examining results stemming from multiple causes. This advantage assists in uncovering the causal complexity between the conditions and outcomes, allowing for an exploration of the simultaneous interactions among various conditions [48]. Furthermore, QCA analysis operates on the logical premise of causal asymmetry, wherein the grouping of paths leading to the emergence of the outcome variable and the absence of the outcome variable exhibit an asymmetric causal relationship. In other words, the condition (or combination of conditions) explaining the existence of the outcome is not necessarily the mirror opposite of the condition (or combination of conditions) responsible for the absence of the same outcome are not mirror opposites [49, 50]. This issue can be further elucidated by analyzing both outcomes.

QCA applications can be categorized into three types: deterministic set (csQCA), multi-valued set (mvQCA), and fuzzy set (fsQCA). In this study, considering the nature of the data, we have chosen fsQCA. FsQCA can handle continuous variables, carve out the attributes of the case with multiple values, and use the set relationship and logical operations rules between sets

to explore the effect of multiple conditions on the results [49]. Additionally, as different combinations of NPD performance element can yield different results, the fsQCA method is well-suited for studying such complex mechanisms of influence.

## Sources of data

The primary data for this study were sourced from the annual report published by the Shenzhen Stock Exchange, the Shanghai Stock Exchange and the Giant Tide Consulting Network. Additionally, some index data were obtained from the Sky Eye. The main focus of this research is on the annual report data published by manufacturing enterprises listed on the main boards of the Shanghai and Shenzhen stock markets from 2000 to 2021.

To ensure the quality and relevance of the study, the initial sample was subjected to the following treatments, following common practices in existing literature: (1) Exclusion of enterprises with significant data deficiencies; (2) Exclusion of companies with abnormal financial statuses designated as ST, *ST, and PT during the sample period; (3) Elimination of enterprises with incomplete data. After this screening and data refinement process, a total of 35 companies were selected for the analysis.

## Measurement and calibration of results and conditions

**Outcome variables.** In this study, NPD performance is serves as the outcome variable with a specific focus on technological innovations created by enterprises. Drawing inspiration from Lahiri's [51], study which suggests that patents can serve as indicators of NPD, we utilize the number of new invention patents granted in 2021 to represent a firm's NPD performance. This patent data was sourced from the Sky Eye website.

**Conditional variables.** The digital transformation capability level divides the indicators into digital infrastructure and digital transformation strategy. Digital infrastructure includes computer hardware, software, telecommunications equipment, etc. [34]. To assess digital infrastructure development, the sum of a company's hardware capital and software capital is chosen. The hardware capital is derived from the ending net value of electronic equipment in the fixed assets section in the enterprise's annual report, and the software capital is determined by the ending net value of software in the intangible assets section [52]. Regarding digital transformation strategy, directly capturing relevant data from a company's annual report can be challenging. Management often discloses their efforts related to "digital transformation" in these reports, which largely reflect the senior management's strategic outlook for both current and future transformations [53]. Therefore, this study uses Python to process the annual report data of the sample enterprises. Employing a digital transformation keyword list compiled by An Tongliang et al. [54] and through Python's Jieba split-word function, the annual report text is analyzed to determine the word frequency of digital transformation-related keywords. This frequency serves as an indicator of the extent of the enterprise's digital transformation strategy.

The research and development input primarily encompass two aspects: R&D personnel input and R&D capital input. R&D investment is seldom quantified using absolute values; rather, it is typically assessed through a relative index—the proportion of research and development input [55]. This proportion is calculated as follows: R&D input/main business income. R&D personnel input is determined by the proportion of R&D personnel relative to the total number of employees within the enterprise [56].

At the level of enterprise heterogeneity, two variables are selected: firm size and firm age. Firm size is quantified based on the number of employees actively engaged in the company's

**Table 1. Calibration results.**

| Research variables | | | Anchor point | | |
|---|---|---|---|---|---|
| | | | **Totally unaffiliated** | **Intersection point** | **Full affiliation** |
| Outcome variable | | NPD performance | 13 | 34 | 95 |
| Conditional variable | Digital transformation capabilities | Digital infrastructure | 8957310.295 | 27734417.11 | 68357283.87 |
| | | Digitalization strategy | 10 | 46 | 80 |
| | Enterprise R&D investment | R&D personnel investment | 5.46 | 14.5 | 18.605 |
| | | R&D capital investment | 3.675 | 6.43 | 9.545 |
| | Firm heterogeneity | Enterprise size | 1932 | 2773 | 5544 |
| | | Enterprise age | 20 | 22 | 23.5 |

operations [57]. Likewise, the ratio of R&D personnel to the total number of employees is used instead [58].

**Variable calibration.** Calibration refers to the assignment of set affiliation to cases [59]. In this paper, we adopt the direct calibration method, as referenced in the work of Fiss [50], Ragin [49], Zhang Ming [48], and Du Yunzhou [59], and others. All the variables in this study are of a fuzzy nature, and the calibration process takes into account the theoretical underpinnings of the research, the distributional characteristics of the case data, and the designation of three anchors for each condition—specifically, the upper quartile, lower quartile, and median of the sample. These anchors are then employed in the calibration process using the algorithm provided by the software. In order to avoid classification difficulties for the set with a calibration value of exactly 0.5, a 0.001 constant was added to all scores to adjust 0.5 to 0.501 [60], and the calibration results are shown in Table 1.

# Results

## Requirements analysis

Before proceeding with fuzzy set truth table analysis, it is necessary to analyze the individual factors. Necessity refers to a condition that must be present for the outcome to occur, but its presence does not guarantee the occurrence of the outcome [61]. Table 2 presents the results of all conditional variables involved in the detection of necessary conditions. It can be observed

**Table 2. Necessary condition test.**

| Conditional variable | NPD high performance | | NPD non-high performance | |
|---|---|---|---|---|
| | **Consistency** | **Degree of coverage** | **Consistency** | **Degree of coverage** |
| Digital infrastructure | 0.735479 | 0.687765 | 0.365007 | 0.37999 |
| ~Digital Infrastructure | 0.336976 | 0.322806 | 0.700076 | 0.746602 |
| Digitalization strategy | 0.688383 | 0.667467 | 0.386159 | 0.416837 |
| ~Digitalization strategy | 0.398563 | 0.36838 | 0.69194 | 0.711982 |
| R&D personnel investment | 0.719237 | 0.684558 | 0.412138 | 0.436699 |
| ~R&D personnel investment | 0.408163 | 0.384113 | 0.7023 | 0.73578 |
| R&D capital investment | 0.65288 | 0.643976 | 0.432151 | 0.47454 |
| ~R&D capital investment | 0.467274 | 0.42501 | 0.675778 | 0.684277 |
| Enterprise size | 0.667311 | 0.649659 | 0.385074 | 0.417352 |
| ~Enterprise size | 0.401522 | 0.369691 | 0.676754 | 0.693685 |
| Enterprise age | 0.551081 | 0.536977 | 0.528799 | 0.573631 |
| ~Enterprise age | 0.562432 | 0.517414 | 0.573164 | 0.587013 |

that the consistency of all the conditions in both the high NPD performance and non-high NPD performance results is lower than 0.9. This suggests that no single condition alone constitutes a necessary condition for the high NPD performance. Consequently, it is essential to analyze the combination of multiple factors in order to study the different paths leading to either high NPD performance or non-high NPD performance.

## Configuration analysis

This paper uses fsQCA3.0 to analyze the necessary parameters. Du Yunzhou et al. [59] noted that the consistency among cases that share the same outcome should exceed the acceptable empirical threshold of 0.85 or 0.8. In this study, the consistency threshold was set to 0.8, the PRI consistency threshold was set to 0.7, and the number of cases threshold was set to 1, based on the inclusion of 35 cases.

During the solving process, three types of solutions are generated: complex solution, intermediate solution and simple solution, depending on the choice of different logical remainder terms. Among them, the complex solution contains the most configuration, the simple solution contain the least, and the intermediate solution strikes a balance with moderate complexity by including logical reminder terms. This paper primary focuses on the intermediate solution, taking the elements that present in both the simple and the intermediate solutions as core conditions, and considering the elements that appear in the intermediate solution as edge conditions.

The assessment of solutions is primarily based on consistency and coverage metrics. Consistency reflects the degree to which a particular solution or all solutions are a subset of the result, while coverage indicate the degree to which the result can be explained.

**A combination of conditions for achieving high NPD performance.** There are five groups of patterns that results in high performance in NPD, as shown in Fig 2. The consistency of individual groupings is 0.90, 0.99, 0.98, 0.91, and 0.90, respectively. These values exceed the generally accepted consistency criterion of 0.80, confirming the validity of the empirical analysis. The overall coverage of the five grouping patterns that led to high NPD performance in the company was 0.60, explaining 60% of the high NPD performance. Among these five groupings, grouping *H1a* exhibits the highest unique and raw coverage, suggesting that grouping *H1a* is the most empirically relevant. Further explanation follows below.

The analysis reveals that grouping *H1a* and *H1b* have the same core condition, i.e., the presence of a high level of R&D investment and firm size plays a central role in improving NPD performance. However, these two groupings have different edge conditions. In grouping *H1a* the presence of digital infrastructure development and digital strategy serves as marginal conditions, playing a secondary role in improving NPD performance. On other hand, in grouping *H1b* the marginal condition consists of the presence of digital infrastructure development and R&D investment share.

Among the other three configurations, configuration *H2* has core conditions that includes digital infrastructure, digital strategy, and R&D investment share, with firm age playing a supporting role. In contrast, R&D staff share and firm size are considered irrelevant conditions in this configuration. Configuration *H3* has core conditions involving digital strategy, R&D staff share and firm age, with the absence of digital infrastructure, R&D investment, and firm size playing a supporting role. Configuration *H4*'s core conditions are the same as those in *H1a* and *H1b*, there are more missing conditions at the edges of this configuration. In this case, the absence of digital infrastructure, digitization strategy, R&D investment ratio, and firm age play a supporting role.

| Prerequisite | High performance in NPD | | | | |
|---|---|---|---|---|---|
| | H1a | H1b | H2 | H3 | H4 |
| Digital infrastructure | ● | ● | ⬤ | ⊗ | ⊗ |
| Digital Transformation | ● | | ⬤ | ⬤ | ⊗ |
| R&D personnel investment | ⬤ | ⬤ | | ⬤ | ⬤ |
| R&D capital investment | | ● | ⬤ | ⊗ | ⊗ |
| Enterprise size | ⬤ | ⬤ | | ⊗ | ⬤ |
| Enterprise age | | | ● | ⬤ | ● |
| consistency | 0.908257 | 0.986553 | 0.981958 | 0.91207 | 0.908257 |
| degree of coverage | 0.346697 | 0.265789 | 0.262891 | 0.0688926 | 0.0597754 |
| Unique coverage | 0.147325 | 0.0633982 | 0.114237 | 0.0290424 | 0.0217365 |
| Consistency of solutions | 0.934566 | | | | |
| Coverage of solutions | 0.595037 | | | | |

**Fig 2. Configurations for achieving high performance in NPD. Note:** ● = core condition present; ⬛ = core condition missing; • = marginal condition present; ⊗ = marginal condition missing, same below.

Based on the core conditions and logical explanations of the five groupings, this paper identifies four driving paths to improve NPD performance: digital innovation-driven, large-scale multi-talent, mature enterprise and digital start-up.

*(1) Digital innovation-driven.* Configuration *H2* suggests that the enterprise's digital transformation plays a significant role in this path. This transformation includes continued investment in digital infrastructure, heightened attention to digitalization, and more comprehensive digital strategy planning. Furthermore, higher R&D investment in companies of a certain age is associated with improved NPD performance. The enhancement of NPD performance is a complex, dynamic process that requires various resources. In this path, the enterprise's digital transformation takes center stage in generating high NPD performance. This is evident in the enterprise's substantial investment in digitization-related software and hardware, along with significant results from digital transformation efforts. Higher keyword frequency related to digital transformation also suggests its positive impact on NPD performance. Thus, digital transformation is crucial antecedent condition for achieving high NPD performance, ensuring that enterprises enhance their NPD performance [21]. However, this path also requires a high proportion of R&D investment by the enterprise, emphasizing that digital transformation cannot exist alone. Enterprises must simultaneously increase R&D investment. Interestingly, enterprise size does not significantly impact the results in this configuration, suggesting that companies can achieve better results by increasing R&D investment alongside digital transformation, regardless of their size. The key takeaway is that a combination of digital transformation and substantial R&D investment is pivotal for driving digital innovation and ultimately enhancing NPD performance.

*(2) Scaled multi-talent type*. Grouping *H1a* and *H1b* share the same core conditions R&D personnel share and firm size. Neither grouping core or marginal missing conditions. These two groupings suggests that large-scale manufacturing firms, maintaining a high R&D staff share can achieve high NPD performance. In larger enterprises with a greater number of employees, a high proportion of R&D personnel ensures a large and high quality workforce, bolstering R&D capabilities, and fostering independent and autonomous innovation. This robust R&D force provides a reliable support for the enterprise's research and development efforts. This is because, for enterprises, R&D personnel represent a critical source of innovation and sustained competitive advantage [62]. The productivity and efficiency of R&D personnel directly influences the conversion efficiency of the enterprise's R&D input and the output of NPD [63].

*(3) Mature and robust*. In Configuration *H3*, the age of the enterprise emerges as a core condition. Under the conditions of digital infrastructure, R&D investment ratio and enterprise size (all marginally missing), a well-established enterprise that strengthens its digitalization strategy and invests in R&D personnel can also achieve high performance in NPD. The interaction among the age of the enterprise, R&D investment and digitalization strategy can significantly contribute to the enterprise's performance. This suggests that mature firms can leverage their stable production and output capacity, along their long-term development capabilities to efficiently allocate resources and excel in their R&D and digitalization endeavors. This enable them to optimize their inputs, enhance the cost-effectiveness of their digitalization strategy and R&D inputs, and ultimately improve NPD performance [55]. Additionally,, mature enterprises have a competitive edge over younger enterprises in terms of resource access, including capital, talent and supply chain advantages.

*(4) Digital start-up*. Configuration *H4* highlights that the core conditions for achieving high NPD performance are a substantial R&D headcount and a large firm size, even with marginal deficiencies in digital strategy, digital infrastructure, and R&D funding. These deficiencies are complemented by the marginal condition of firm age. This grouping underscores that for large-scale multi-talent firms, increasing the proportion of R&D personnel can enhance their NPD competitiveness and performance. This holds true even when digital transformation efforts are either at an early stage or have been implemented to a limited extent within the company.

**A combination of conditions for achieving non-high NPD performance.**   This study also analyzed the scenarios non-high NPD performance, as shown in Fig 3. The analysis reveals that the patterns associated with of non-high NPD performance are asymmetric compared to those leading to high NPD performance. There are also five group paths. Both *L1a* and *L1b* highlight that inefficiencies in digitalization input, innovation activity input, and the enterprise's nature will inevitably result in non-high NPD performance. When any two of the three groups of conditions, *L2*, *L3*, and *L4*, are absent, even if the core conditions of remaining group exists, non-high performance is inevitable. Configuration *L3*, in particular, illustrate that enterprises that undertake digital transformation without careful consideration of their own circumstances and blindly increase their investment ins research and development personnel and capital may encounter the "digital paradox" phenomenon [20]. This phenomenon signifies that the enterprise's digitalization efforts do not yield the expected benefits [63].

## Robustness analysis

There are four common methods for conducting robustness testing, which include raising the case consistency threshold, improving PRI consistency, adding or deleting cases, and

| Prerequisite | NPD non-high performance | | | | |
|---|---|---|---|---|---|
| | **L1a** | **L1b** | **L2** | **L3** | **L4** |
| Digital infrastructure | ⊗ | ⊗ | ⊗ | ⊗ | ⊗ |
| Digitalization strategy | ⊗ | ⊗ | ⊗ | ● | • |
| R&D personnel investment | ⊗ | ⊗ | | ● | ⊗ |
| R&D capital investment | | ⊗ | ⊗ | • | • |
| Enterprise size | ⊗ | | ⊗ | ⊗ | • |
| Enterprise age | ⊗ | ⊗ | ● | ⊗ | ● |
| consistency | 0.922771 | 0.976182 | 0.964365 | 0.973262 | 0.869919 |
| degree of coverage | 0.239126 | 0.155603 | 0.234841 | 0.0987092 | 0.0580323 |
| Unique coverage | 0.112268 | 0.0477275 | 0.185541 | 0.0542358 | 0.0385074 |
| Consistency of solutions | 0.940633 | | | | |
| Coverage of solutions | 0.5749 | | | | |

**Fig 3. Configurations for achieving non-high performance in product development. Note:** • = core condition present; ⊗ = core condition missing; • = marginal condition present; ⊗ = marginal condition missing, same below.

introducing additional conditions. Any of these methods can be selected for robustness testing [48]. In this paper, we choose to improve the original consistency of the test, the original consistency threshold from 0.8 to 0.85, the use of a more stringent threshold to start the analysis, the results are shown in Fig 4. Overall the consistency of the overall solution before and after adjusting the threshold is improved from 0.93 to 0.97, and the coverage is reduced from 0.60 to 0.55.

Upon comparing the grouping results before and after adjustment, it is evident that there are still five groupings presents after the threshold adjustment. Groupings *H2* and *H3* remain entirely consistent with their previous configurations. *H4* and *H2* maintain the same paths as before, with only slight changes to the edge and core conditions. The *H1a* path is a subset of the previous paths. This demonstrate the robustness of the findings presented in this paper.

## Conclusions and implications

### Conclusions

This study focuses on the annual report data published by listed manufacturing enterprises on the main boards of Shanghai and Shenzhen in 2021. Grounded in resource orchestration theory and the capability-resource-structure framework, it selects six antecedent variables from the three dimensions: digital transformation capability, enterprise R&D investment, and enterprise heterogeneity. A cohort analysis is conducted to assess the impact of different combinations of variables on NPD performance, leading to the following key conclusions:

| Prerequisite | High performance in NPD after improving consistency | | | | |
|---|---|---|---|---|---|
| | H1a | H1b | H2 | H3 | H4 |
| Digital infrastructure | • | • | ● | ⊗ | ⊗ |
| Digital Transformation | ● | | ● | ● | ⊗ |
| R&D personnel investment | ● | • | | ● | ● |
| R&D capital investment | | • | ● | ⊗ | ⊗ |
| Enterprise size | • | • | | ⊗ | ● |
| Enterprise age | • | | • | ● | • |
| consistency | 0.952863 | 0.986553 | 0.981958 | 0.91207 | 0.908257 |
| degree of coverage | 0.244113 | 0.265789 | 0.262891 | 0.0688926 | 0.0597754 |
| Unique coverage | 0.0996861 | 0.118343 | 0.114237 | 0.0290424 | 0.0217365 |
| Consistency of solutions | 0.966937 | | | | |
| Coverage of solutions | 0.547398 | | | | |

**Fig 4. Grouping generated by elevated case consistency thresholds. Note:** ● = core condition present; ⊗ = core condition missing; • = marginal condition present; ⊗ = marginal condition missing, same below.

1. The primary paths for companies to improve NPD performance includes digital innovation-driven, scaled multi-talent, mature and robust, and digital start-up. There exists an asymmetric causal relationship between the factors that comprising these four paths and performance.

2. None of the six antecedent variables within the three dimensions of digital transformation capability, enterprise R&D investment and enterprise heterogeneity independently constitutes a necessary condition for achieving high NPD performance. A single condition has limited explanatory power for high NPD performance, and at least two dimensions of factors need to work in synergy to generate NPD performance.

3. When comparing the high-performance paths of NPD with the non-high-performance paths, it is observed that enterprise heterogeneity level factors are core and consistently present in most high-performance paths, while they are frequently missing in non-high-performance paths. This underscores the significance of enterprise heterogeneity in influencing outcomes. It indicates that enterprises should thoroughly consider their current circumstances and characteristics during digital transformation, make judicious use of resources, and avoid blind investments.

4. Scale-ups can improve their NPD performance by increasing their investment in R&D personnel during the initial stages of digital transformation, particularly when digital coverage is low.

## Limitations and future directions

The results of this study exhibit strong consistency and offer robust interpretation. However, there is a limitation in terms of coverage. With only 35 cases selected, the resulting grouping can only account for certain paths. This suggest a potential omission in the selection of antecedent variables, emphasizing the need for future research to explore more powerful antecedent variables to enhance result validation.

Additionally, due to data availability constraints, this paper analyzed cross-section data from manufacturing enterprises in 2021. It did not account for dynamic changes and data lag. A future research direction could involve analyzing enterprise NPD performance in correlation with the incremental stages of digital transformation to provide a more comprehensive understanding.

## Supporting information

**S1 Data.**
(XLSX)

## Author Contributions

**Conceptualization:** Daohai Zhang.

**Data curation:** Daohai Zhang, Xianqiao Hou, Jiayuan Guo.

**Formal analysis:** Daohai Zhang.

**Funding acquisition:** Daohai Zhang.

**Investigation:** Daohai Zhang, Xianqiao Hou.

**Methodology:** Daohai Zhang, Jiayuan Guo.

**Project administration:** Daohai Zhang, Jiayuan Guo.

**Resources:** Daohai Zhang, Jiayuan Guo.

**Software:** Xianqiao Hou.

**Supervision:** Xianqiao Hou.

**Validation:** Daohai Zhang, Xianqiao Hou, Jiayuan Guo.

**Visualization:** Xianqiao Hou.

**Writing – original draft:** Daohai Zhang, Xianqiao Hou.

**Writing – review & editing:** Daohai Zhang, Xianqiao Hou.

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
