## [Decision Letter · Decision Letter 0]

27 Sep 2023

PONE-D-23-28468How does digital transformation improve new product development performance from the perspective of resource orchestration? -- Analysis based on configurationPLOS ONE

Dear Dr. Zhang,

Thank you for submitting your manuscript to PLOS ONE. After careful consideration, we feel that it has merit but does not fully meet PLOS ONE’s publication criteria as it currently stands. Therefore, we invite you to submit a revised version of the manuscript that addresses the points raised during the review process.

We look forward to receiving your revised manuscript.

Kind regards,

Han Lin

Academic Editor

PLOS ONE

Journal Requirements:

Additional Editor Comments:

It might be helpful with new citations of up-to-date and high-quality papers on the subjects covered in manuscript.

Also, the author(s) should carefully format the manuscript following the guidance of the journal, such as the headers, references, and so on. There are mistakes in the citation (reference), such as missing volume, issue and page.

The authors are recommended to examine and proof the language. In addition, there are some typos and grammatical errors that need further attention.

Reviewers' comments:

Reviewer's Responses to Questions

**Comments to the Author**

1. Is the manuscript technically sound, and do the data support the conclusions?

Reviewer #1: Yes

Reviewer #2: Yes

Reviewer #3: Yes

2. Has the statistical analysis been performed appropriately and rigorously? 

Reviewer #1: Yes

Reviewer #2: Yes

Reviewer #3: Yes

3. Have the authors made all data underlying the findings in their manuscript fully available?

Reviewer #1: Yes

Reviewer #2: Yes

Reviewer #3: Yes

4. Is the manuscript presented in an intelligible fashion and written in standard English?

Reviewer #1: Yes

Reviewer #2: Yes

Reviewer #3: Yes

5. Review Comments to the Author

Reviewer #1: In this paper, based on the 2021 financial data of 35 companies operating in China's Shanghai and Shenzhen stock markets, the author studies the factors affecting the new product development performance of Chinese enterprises under the background of digital transformation, and makes an empirical analysis, and draws some interesting and useful conclusions. Including which digital transformation factors can better promote the performance of enterprise new product development, and what is its internal mechanism. Personally, the research data of this paper is detailed and accurate, the method is appropriate, and the relevant conclusions have good practical application value.

Reviewer #2: 1 The sample size of the article is 35, why was 35 samples chosen and on what basis

2 The contribution of the text is not sufficiently summarized in the conclusion

3 The article does not present the innovation of this paper, the innovation is not clear, need to focus on summarizing the innovation

Reviewer #3: The topic is interesting and original, but I feel the paper has a number of shortcomings that do not allow its publication, at least at the current level of development. The issues are listed in the following:

1 The article has different expressions for the same term, for example, "capacity of digital transformation" and "digital transformation capabilities", please standardize the terminology.

2 Whether the number of cases in this paper is selected based on the evidence.

3 What is the innovation point of this paper and what is the difference between this paper and similar literature?

4 Page 12. 2.1 Digital transformation capabilities (Capability Layer) -> 2.1 Digital transformation capabilities ,

5 Page 13. 2.2 Enterprise R&D inputs (Resource Layer) ->Enterprises’ R&D Input

6 Page 14. 2.3 Firm heterogeneity (Structural Layer

6. PLOS authors have the option to publish the peer review history of their article (what does this mean?). If published, this will include your full peer review and any attached files.

Reviewer #1: No

Reviewer #2: No

Reviewer #3: No

---

## [Author Response · Author response to Decision Letter 0]

3 Oct 2023

Dear reviewer,

We sincerely thank the editor and all reviewers for their valuable feedback that we have used to improve the quality of our manuscript. As you are concerned, there are several problems that need to be addressed. According to your nice suggestions, we have made extensive corrections to our previous draft, the detailed corrections are listed below:

Reviewer #1:

In this paper, based on the 2021 financial data of 35 companies operating in China's Shanghai and Shenzhen stock markets, the author studies the factors affecting the new product development performance of Chinese enterprises under the background of digital transformation, and makes an empirical analysis, and draws some interesting and useful conclusions. Including which digital transformation factors can better promote the performance of enterprise new product development, and what is its internal mechanism. Personally, the research data of this paper is detailed and accurate, the method is appropriate, and the relevant conclusions have good practical application value.

The author’s answer:

Many thanks to the reviewers for recognizing this study.

Reviewer #2: 

1 The sample size of the article is 35, why was 35 samples chosen and on what basis

2 The contribution of the text is not sufficiently summarized in the conclusion

3 The article does not present the innovation of this paper, the innovation is not clear, need to focus on summarizing the innovation

The author’s answer:

1 The complementary strengths of qualitative and quantitative analyses are integrated in the QCA analytic approach, which is suitable for both small case-count studies (10 or less than 15 cases), medium-sized samples (10 or 15 to 50 case-counts), and large samples exceeding 100 case-counts (Ragin, 2008; Berg-Schlosser et al., 2009; Crilly et al. Crilly et al., 2012; Greckhamer, 2015; Greckhamer, Misangyi & Fiss, 2013), as well as studies with large samples of cases (Fiss, 2011; Misangyi & Acharya, 2014). For medium-sample studies, the ideal number of conditions is generally between 4 and 7 (Berg-Schlosser & De Meur, 2009); for large-sample studies, the number of conditions can be higher (Misangyi & Acharya, 2014) 

Therefore, this paper has six antecedent variables, which is suitable for medium-sized samples, so the cases were selected and the firms with incomplete data were excluded, and the final number of cases was determined to be 35.

2 The contribution of this article is repeated in the introduction and conclusion sections, and the article contribution in the introduction has been revised and the article contribution in the conclusion section has been deleted. The revised article contribution is below:

The paper's contributions are threefold. Firstly, by adopting a configuration perspective and focusing on a country listed manufacturing enterprises, this study delves into the impact of six conditional combinations on NPD performance at three levels: ability, resources, and structure. This approach enhances our comprehension of the intricate interactions among multiple factors that influence NPD performance. Additionally, it offers valuable insights for deepening our understanding of the complexities behind NPD performance. Moreover, it provides valuable insights for driving the development of China's manufacturing industry and enhancing enterprise competitiveness. Secondly, by incorporating the heterogeneous nature of enterprises into the research framework, this paper enriches the study of enterprise performance scenarios. This enrichment contributes to elevating digital capabilities, enhancing NPD abilities, and boosting the competitive advantages of various enterprises. Thirdly, the data of this study are all from the annual reports of enterprises, which are real and credible, and the analysis results are highly persuasive. This study not only helps enterprises to better understand the mechanism of resource management effectiveness in the new product development process, but also facilitates enterprise managers to formulate effective resource management plans to improve their new product development performance.

3 The innovations of this paper are combined in the contribution of the article, of which the most central innovation is: taking domestic manufacturing enterprises as the research object, for the first time in the Chinese context, based on the theory of resource orchestration, we analyze the mechanism of the three-tier resource orchestration affecting the creativity of new products, and combined with the grouping perspective, we study the role of the digital transformation capability, R&D investment and enterprise heterogeneity on the impact of the NPD , as well as the path of the impact. Combining the resource orchestration theory and the group theory, this paper innovatively obtains the results of the combination conditions affecting the performance of NPD based on the "capability-resource-structure" theoretical model, which is more applicable than the single-variable research, and at the same time, the group theory can study the non-causal symmetric results, and the influence path of low performance is also studied. The research is also carried out on the path of low performance.

Reviewer #3: 

The topic is interesting and original, but I feel the paper has a number of shortcomings that do not allow its publication, at least at the current level of development. The issues are listed in the following:

1 The article has different expressions for the same term, for example, "capacity of digital transformation" and "digital transformation capabilities", please standardize the terminology.

2 Whether the number of cases in this paper is selected based on the evidence.

3 What is the innovation point of this paper and what is the difference between this paper and similar literature?

The author’s answer:

1 Thanks for the correction, this issue has been revised in the manuscript.

2 The issue is the same as the point made by the previous reviewer, which is explained above.

3 The innovation of this paper is that it takes domestic manufacturing enterprises as the research object, and analyzes the mechanism of three-level resource allocation affecting new product creativity based on resource orchestration theory for the first time in the Chinese context, and investigates the influence of digital transformation capability, R&D investment and enterprise heterogeneity on NPD and the influence path by combining the group perspective. Combining the resource orchestration theory and the group theory, this paper innovatively obtains the results of the combination conditions affecting the performance of NPD based on the "capability-resource-structure" theoretical model, which is more applicable than the single-variable research, and at the same time, the group theory can study the non-causal symmetric results, and the influence path of low performance is also studied. is also studied. Compared with a large number of empirical papers in this field, this paper adopts data from annual reports of enterprises, which is real and effective, and the analysis results are more persuasive.

All formatting and typo issues raised by other editors have been corrected in the manuscript.

Thank you very much for your attention and time. Look forward to hearing from you.

Yours sincerely:

Daohai Zhang

03 Oct,2023

---

## [Decision Letter · Decision Letter 1]

31 Oct 2023

How does digital transformation improve new product development performance from the perspective of resource orchestration? -- Analysis based on configuration

PONE-D-23-28468R1

Dear Dr. Zhang,

We’re pleased to inform you that your manuscript has been judged scientifically suitable for publication and will be formally accepted for publication once it meets all outstanding technical requirements.

Kind regards,

Han Lin

Academic Editor

PLOS ONE

Additional Editor Comments (optional):

Reviewers' comments:

Reviewer's Responses to Questions

**Comments to the Author**

1. If the authors have adequately addressed your comments raised in a previous round of review and you feel that this manuscript is now acceptable for publication, you may indicate that here to bypass the “Comments to the Author” section, enter your conflict of interest statement in the “Confidential to Editor” section, and submit your "Accept" recommendation.

Reviewer #2: All comments have been addressed

Reviewer #3: All comments have been addressed

2. Is the manuscript technically sound, and do the data support the conclusions?

Reviewer #2: Yes

Reviewer #3: Yes

3. Has the statistical analysis been performed appropriately and rigorously? 

Reviewer #2: Yes

Reviewer #3: Yes

4. Have the authors made all data underlying the findings in their manuscript fully available?

Reviewer #2: Yes

Reviewer #3: Yes

5. Is the manuscript presented in an intelligible fashion and written in standard English?

Reviewer #2: Yes

Reviewer #3: Yes

6. Review Comments to the Author

Reviewer #2: (No Response)

Reviewer #3: It is an interesting study. In summary, I think that this paper is suitable to be published in PLOS ONE.

7. PLOS authors have the option to publish the peer review history of their article (what does this mean?). If published, this will include your full peer review and any attached files.

Reviewer #2: No

Reviewer #3: No

---

## [Editor Report · Acceptance letter]

17 Nov 2023

PONE-D-23-28468R1 

How does digital transformation improve new product development performance from the perspective of resource orchestration? -- Analysis based on configuration 

Dear Dr. Zhang:

I'm pleased to inform you that your manuscript has been deemed suitable for publication in PLOS ONE. Congratulations! Your manuscript is now with our production department. 

Kind regards, 

on behalf of

Dr. Han Lin 

Academic Editor

PLOS ONE